# Morality, Voluntary Laws, and State Neutrality

**Yitzhak Benbaji**

Law Faculty, Tel Aviv University, Tel Aviv 6997801, Israel; ybenbaji@gmail.com

**Abstract:** Kantian *political* philosophies stress that a state ought to be "neutral" (Rawls), "minimal" (Nozick), or "public" (Ripstein's Kant), as part of its duty to respect its citizens' freedom to pursue whatever ends these citizens find valuable. States are under duty merely to secure citizens' independence from each other and from the state. In contrast, Kantian *morality* contends that individuals are subject to a duty to pursue certain "obligatory" ends, viz., ends that emerge from the intrinsic value of personhood and autonomy. In some cases, hindering one's freedom is necessary for promoting these ends. This essay describes circumstances in which a *legal* right to interfere with one's property and body in promoting obligatory ends is justified, even though such a right compromises states' neutrality. This description sheds a new light on the relation between the optimal legal system ("Right") and morality ("Virtue") and between justice and truth.

**Keywords:** Kant's political philosophy; Arthur Ripstein; justice as fairness; John Rawls; political liberalism; democracy

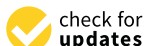



## 1. Introduction

Kantian (non-consequentialist) morality and Kantian political philosophy—as thinkers such as John Rawls (Rawls 1971, [1993] 2005, 2001), Robert Nozick[1], and Arthur Ripstein's Kant (Ripstein 2009) develop it—employ two seemingly similar but actually very different concepts of freedom.[2] Kantian political philosophies imply that a state ought to be "neutral" (Rawls)[3], "minimal" (Nozick 1974, pp. 26–28), or merely "public" (Ripstein's Kant)[4], as part of its duty to treat its citizens as free and equal. Indeed, despite the differences between them, Rawls, Nozick, and Ripstein all maintain that as far as citizens do not hinder other citizens' freedom, the state ought to empower them to pursue the ends that they set themselves without attending to the life plans to which others are committed and to the ends that they pursue. The state should identify, secure, and fairly distribute the things that constitute the most indispensable means to any system of human ends whatever its particular content.[5] It follows that citizens' rights to "political" or "external" freedom

---

1   (Nozick 1974, pp. 161–62). While Nozick's inspiration is John Locke, Kant's categorical imperative is the basis of his understanding of rights as side-constraints. See (ibid., pp. 30–35).

2   The distinction resembles but is *not* identical to Isaiah Berlin's. See his "Two Concepts of Liberty" (1958) reprinted in his *Liberty*, (Berlin 2002).

3   I understand Rawls' commitment to "neutrality" as a rejection of comprehensive liberalism which Rawls (mistakenly?) attributes to Kant and Mill. Rawls maintains that comprehensive liberalism "may lead to requirements designed to foster the values of autonomy and individuality as ideals to govern much if not all of life" (Rawls [1993] 2005, p. 199) in a way that necessitates a systematic use of "oppressive use of state power" (p. 37). See also (Rawls [1993] 2005, p. 191) and Rawls' restriction of neutrality to "constitutional essentials" (p. 234). My understanding of the ideal of neutrality is close to the one Jonathan Quong offers in his "Liberalism Without Perfectionism" (Quong 2011), and to Allan Patten's reinterpretation of this ideal, in his *Equal Recognition* (Patten 2014, pp. 104–37).

4   "A state is required to act for public purposes, but prohibited from acting for private ones, and individual rights constrain the means that the state may use in pursuit of public purposes" (Ripstein 2009, p. 208).

5   I put this theme in Rawlsian terms. See (Rawls 1971, p. 93). Nozick and Ripstein would use different terms to express a similar view. Their disagreements notwithstanding, they all agree that liberties and private property ought to be protected by the state, partly because liberties and property are essential to legitimate human purposeful agency. Despite appearances to the contrary, distributive considerations must enter Ripstein's

---

correlates with the states' duty to be neutral regarding the ends that they pursue and their life plans.

There is a contrast between Kantian political freedom and Kantian moral freedom. Note first, that clearly, Kantian non-consequentialism does interfere with our life plans. Kantian morality contends that individuals are subject to a duty to pursue certain "obligatory" ends, viz., ends that emerge from the intrinsic value of personhood and autonomy. Kantians insist, for example, that the view of "religious deniers", who believe that only God is intrinsically valuable, and that humans ought to be His slaves, are both mistaken and defective. Instead, persons are entitled to respect thanks to their personhood, and any action that compromises their autonomy is pro tanto objectionable.[6] The general duty to respect one's and others' personhood implies a more particular duty that will concern me in this essay: the duty to use the things that one owns in rescuing a stranger from death, if the cost of the rescue is not excessive. Rather than compromising *moral* freedom, the duty to pursue this and other obligatory ends is a manifestation of its intrinsic value.

The tension is well known. The existence of obligatory ends notwithstanding, Kantian political philosophy implies that, typically, forcing people to pursue obligatory ends (by, say, sanctioning their failure to do so) would violate liberalism's presumption against oppression. Particularly, religious deniers have a *moral* right that the state will not force them to pursue the ends that Kantian morality renders obligatory. Kantian political philosophies imply, then, that the optimal law requires merely to respect the *external* or *political* freedom of others, rather than their *moral freedom* or *personhood* (See Ripstein 2009, p. 52). While Kantian political philosophers differ as to the scope of the liberty-right against state interference with one's body and property, they agree that where the interferer's reason for hindering one's freedom is grounded in the comprehensive doctrine to which citizens are committed, the interference in question is pro tanto objectionable. Coercively preventing a person from interfering with another person's body and property is one of the state's fundamental duties; forcing a person to pursue an end because a comprehensive doctrine renders it obligatory is pro tanto objectionable.

There is, then, a clear sense in which Kantians such as Rawls, Nozick, and Ripstein are *separationists*. Separationism attaches intrinsic value to both personhood and external freedom. To use Ripstein's terms, these philosophers separate "Right" from "Virtue": while we are under a duty *of virtue* to pursue obligatory ends, in respecting our freedom as independence, the legal system should entitle us to violate this moral duty (Ripstein 2009, pp. 12 and 268–69). The optimal legal system should merely force private persons to fulfill their duty *of right* to respect others' freedom by protecting their body and property. Put in terms introduced in Rawls' political liberalism, the *true* comprehensive doctrine grounds duties of virtue to which all are subject. The vindication of such duties is "metaphysical". By contrast, duties of right are political, not metaphysical. *Reasonable* comprehensive doctrines would support these political rights, since they all respect—each for its own reason—the political right of individuals to choose their life plan, based on their (true or false) conceptions of the good.[7]

To prevent misunderstanding, let me emphasize that separationism is not meant to be a reading of Kant's own practical philosophy. Rather, I take it to be general a feature shared

---

and Nozick's picture. After all, they must be concerned with fairly distributing the costs of sustaining the minimal/public state.

[6] For an insightful discussion see (Rawls 2000, at p. 177 and especially, at pp. 187 ff.) where he argues that one is under a duty to promote the freedom, the talents, the proper interests, and the happiness of others. In *Anarchy, State and Utopia*, moral duties that do not correlate with rights manifest themselves in Nozick's discussion of the proper attitude to animals. For the relation between Right (viz., the optimal law) and Kant's categorical imperative, see (Ripstein 2009, pp. 355–88).

[7] Thomas Pogge suggested that Kant's political philosophy is an early version of political liberalism in his (Pogge 2002, pp. 153–58). Both Ripstein and Pogge insist that while autonomy, intrinsic value, right action, and rightness are central to understanding Kant's moral philosophy, political philosophy centralizes a distinct value. As I will read it, Rawls' (later) political liberalism is another version of this separationist move.

by very *different* Kantian practical philosophies (viz., Rawls', Nozick's, and Ripstein's; only the latter is presented as an interpretation of Kant).

Separationism faces a challenge, however. Many actual legal systems confer on third parties a *legal* right to interfere with a person's property and body on this person's behalf. Moreover, they confer a legal right to interfere with a person's property and body on another person's behalf, in cases of necessity and duress. In other words, the law allows third parties to protect personhood, even if this involves interfering with the property of a religious denier.

Without refuting other possible solutions to this problem, this essay construes the relation between non-consequentialist morality ("Virtue") and the morally optimal legal system ("Right") in a noble way, which I call "*reduced* separationism". According to reduced separationism, citizens ought to establish and support a neutral legal system that determines and protects external freedom. Indeed, individuals can be permissibly *forced* to do so. Yet, citizens should also *voluntarily* accept a legal system that limits the reach of their external freedom in circumstances in which interfering with their body and property is necessary for promoting (certain) obligatory ends. Crucially, though, they have a right against being forced to accept such redefinition of their external freedom. Reduced separationism draws a distinction between peremptory and voluntary legal rules and assumes that the acceptance of the latter is morally effective. By freely accepting legal rules that allow interference with our property and body in circumstances in which this is a necessary means for advancing certain obligatory ends, we voluntarily waive our right against the interference in question.

I proceed as follows. Sections 2 and 3 present in more detail a non-consequentialist distinction between duties of right (grounded in the political value of external freedom) and duties of virtue (grounded in the moral value of humanity or moral freedom, according to Kantian metaphysics). Section 4 shows that the tension between them is deeper than separationists acknowledge. Section 5 develops the theory that this essay advocates, "reduced separationism". Section 6 concludes.

## 2. Neutrality and Non-Consequentialist Separationism

As Arthur Ripstein reads it, Kant's political philosophy is based on a single idea, the idea of freedom as independence. A person is sovereign over herself—her body, mind, and property—and, as such, possesses a right against being subordinated to another person. Based on this idea, the Universal Principle of Right commands that the freedom of one person—as it is manifested in his or her rational purposiveness—would be consistent with the rational purposiveness of others.

The normative powers of the state are all related to this principle: the state should determine the scope of citizens' self-mastery and protect it. The legal system of a justified state determines who owns what and, in doing so, enables citizens to respect the freedom of others, and empower them to pursue the ends that they set out for themselves, without attending to the ends that others pursue. Additionally, the state assures each person that so far as she does not violate the bodily and property rights of other private persons, her body and property will not be subject to others' choices (Ripstein 2009, p. 14). According to Ripstein, the Kantian state is minimal, in an important sense: it has no normative power to advance any purpose that cannot be justified in terms of external freedom.

Similar ideas appear in most non-consequentialist political philosophies. Libertarians like Nozick centralize the idea of self-ownership. The only justifiable state is the minimal state that protects self-ownership, viz., a state that is limited to the protection of the rights of person, property, and contract. The minimal state is a kind of protective agency that enforces conflict-diminishing agreements and resolution procedures (Nozick 1974, pp. 48–53). Its role is to protect the rights of people with conflicting interests, conflicting ends, and incompatible comprehensive doctrines and life plans.

In securing state neutrality, Rawls' egalitarianism prioritizes the first principle of justice, which determines and secures citizens' basic liberties, viz., those liberties that

enable them to pursue, modify, and abandon their (reasonable) life plans, whatever they might be. Any use of force should be acceptable from the perspective of all reasonable comprehensive doctrines. The basic structure of society governs religious deniers as well as "hedonistic deniers", who believe that pleasure and welfare are the only things which have intrinsic value. They abruptly reject the Kantian idea that rationality and autonomy have intrinsic value. Deniers (of both types) would resent a limitation of their freedom grounded in the intrinsic value of personhood just as secular humanists would oppose any use of state force justified by the intrinsic value of welfare or of God. A state ought not to prefer one of these views. Accordingly, the state ought to avoid forcing "requirements designed to foster the values of autonomy and individuality as ideals to govern much if not all of life" (Rawls [1993] 2005, p. 199; quoted in Pogge 2002, p. 154).

　　Thus, notwithstanding the deep differences between the theories elaborated by Rawls, Nozick, and Ripstein, they all imply that the law ought to ensure that citizens can respect the bodily and property rights of others. Additionally, it should empower them to pursue the ends they choose by using the means to which they are entitled, without taking others into account and without being subject to others' choices.

　　Put differently, all three thinkers develop versions of political liberalism and strongly oppose "comprehensive liberalism". They hold that in shaping its policy the liberal state should not appeal to "conceptions of what is of value in human life", and to "ideals of personal virtue and character that are to inform much of our non-political conduct" (Rawls, ibid., at p. 175; quoted in Pogge ibid.). Comprehensive liberalism is an illiberal political philosophy, since it can be maintained only by violating "its own principles against the use of state oppression" (Rawls, ibid., at p. 37, n. 39; quoted in Pogge ibid.). Indeed, the neutrality of the state enables it to "stand above the potentially contested . . . and shifting interests that persons may contingently attribute to themselves and to others." (Ibid., p. 149). In particular, secular humanists, religious, and hedonistic deniers would support a regime that ensures their capacity to pursue their life plans. Humanists and deniers, each for their own reason, acknowledge that citizens possess a moral right to adopt a life plan that might look to others metaphysically inadequate and morally defective. Both stress that citizens possess a moral right that the state will not force a metaphysically adequate life plan on them.[8]

　　Of course, in Rawls', Nozick's, and Ripstein's Kantian views, states are entitled to interfere with the property and body of their citizens in advancing ends that these citizens do not share. States are under a duty to protect the freedom of their citizens, and accordingly, are entitled to the necessary means that enable them to do so. States should establish police and military forces; accordingly, they are entitled to promote these ends by fairly taxing their citizens. Kantians would nevertheless insist that states' interventions should *not* be justified by appealing to any specific comprehensive doctrine, or any specific set of ends that some citizens pursue.[9]

### 3. On the Right/Virtue Relation: The Practical Angle

　　How is Kantian morality (Virtue) related to Kantian political neutrality (Right) *in practice*? How is Kantian morality related to the law that Kantian political philosophy obligates states to positivize?[10]

　　The first thing to note is a significant overlap between these normative landscapes. Typically, violating one's self-mastery—the value that the legal system should protect—is not only a violation of a duty of right but also a violation of a duty of virtue, viz., a failure to

---

8　Here is Ripstein's Kant: "Each of the state's internal duties restricts its ability to act through its officials to those acts consistent with the rights of its citizens. Its only end, then, is to observe the restrictions presupposed by its basic mandate; its positive provision of, for example, public roads or support for the poor is just the restriction of its other activities to terms consistent with right. The only questions it faces are questions of how to give effect to a rightful condition" (Ripstein 2009, p. 204).

9　See (Pippin 2007, pp. 416–46), where these attempts are discussed (and rejected). The derivationist reading of the Right/Ethics distinction is elaborated in (Guyer 2002, pp. 23–62).

10　This section relies on (Ripstein 2009, pp. 355–89).

treat personhood respectfully. Indeed, besides being illegal, and besides being a violation of the duty to respect the external freedom of others, negligibly destroying or using things owned by others (without their authorization) would be a violation of a fundamental duty of virtue. It is morally impermissible to pursue a permissible end by using others' humanity as a means; intentionally using the things that others own or negligibly injuring them is typically a violation of this moral prohibition.

Yet, the overlap between Virtue and Right is only partial. First, Kantian morality prohibits disrespectful *self*-treatment. A person should not use herself as a means for producing pleasure. As we shall immediately see, an important strand in non-consequentialism renders suicide whose aim is to end bearable pains impermissible. Yet, a neutral legal system regulates only "the outer relations" between citizens; it is not concerned with one's own life plan.

Consider another phenomenon that manifests the restricted scope of the optimal law. Unlike a legally regulated contract, an informal promise does not include transference of a legally protected interest from the promisor to the promisee. Such a promise has no effect on the scope of the promisor's property rights, say. Hence, there is no duty of right to keep the promise—whereas, obviously, according to any Kantian morality, it is a duty of virtue to uphold any promise, formal and informal alike.[11] The centrality of the misfeasance/nonfeasance distinction in the theory of private law conveys the same idea: when you use or destroy the things that belong to another person without her permission you violate her self-mastery, whereas in failing to aid her, you merely fail to do something for her. This failure involves no violation of a legal and moral Right-based claim that she holds against you: "[t]he person who declines to exercise his own self-mastery in aid of your wishes or needs does not thereby become your master." (Ripstein 2009, p. 45). While being an instance of non-feasance, such a failure often involves a violation of a duty of virtue.

Thirdly, the converse direction is also possible. In some cases, the Right-based law would render certain actions illegal while most Kantian comprehensive doctrines would not condemn them as immoral. A trespass against land involves using another person's land without the owner's authorization. Now, imagine a person who *mistakenly* interferes with another's land. The law would imply that this person is a trespasser even if she uses land that she mistakenly thought to be her own. Most Kantians would insist, however, that she violates only a (legal) duty of right, not a duty of virtue to respect others' external freedom or personhood (Ripstein 2009, p. 381).

These phenomena do not resolve the debate between separationism (according to which Right and Virtue are irreducible to each other) and its opponents who aim to derive Right from Virtue. To the contrary, they might convey the impression that Virtue and Right are elements in one harmonious normative system, since, typically, duties of right are coercible duties of virtue. Indeed, the fundamental aspect of the relation between Right and Virtue has to do with states' normative power to use force. The state is entitled to enforce the optimal law, by coercively interfering with citizens' body and property. On the other hand, according to Kantians including Ripstein, Nozick, and perhaps Rawls, even when a person ought to aid a stranger, she has a moral right that others—the state included—not force her to do so. It might be thought, then, that duties of right are, in fact, enforceable duties of virtue.

I believe (but will not be able to offer an elaborated argument for this belief) that the impression that Right and Virtue (as conceived in Kantian philosophies discussed above) are two elements in one normative system is delusional. Kantian political morality requires neutrality: within certain limits, the optimal law should enable citizens to pursue the ends they set themselves whatever these ends might be; private citizens are under legal and moral duty to respect its rulings. In sharp contrast, Kantian morality introduces duties

---

[11]    See (Pippin 2007, p. 421): "Not all broken promises or lies are violations of right."

of virtue to pursue obligatory ends.[12] The value of humanity and rationality implies that there are objectives that are "valid for all reasonable and rational persons in the sense that every such person must count these ends as ends they are to advance." (Rawls 2000, p. 194). And, as we shall immediately see, there might be circumstances in which in promoting an obligatory end one must violate the self-mastery of others. I will assume, then, the Kantian practical philosophy is separationist in a strong sense. The intrinsic values of external freedom and neutrality are irreducible to the value of personhood; they often compete with each other.[13] This is why separationists need to stress that Right is *morally* significant, viz., that "all duties of right are 'indirectly' duties of virtue ... there is an obligation of virtue to act on the principles of right, to make them your own principles of action (Ripstein 2009, p. 389), and that, "[e]veryone has the end of respecting the rights of justice [i.e., Right], for this too is meritorious and an obligation ... of virtue ... " (Rawls 2000, p. 209).

Let me summarize separationism in three propositions:

*Separationism:*

*Virtue1.* Individuals are under duty to acknowledge and to respect the value of personhood. Specifically, they are under pro tanto duty not to compromise this value by using the humanity in themselves and others as a means, and they are under duty to enhance this value by pursuing obligatory ends.

*Right.* The state ought to determine the scope of the bodily and property rights of its citizens and to protect them.

The state is entitled to hinder private freedom *only* for advancing *its* obligatory ends as a public entity, viz., securing freedom and just distribution of the means to realize it. It must be neutral with respect to the other ends that its citizens pursue.

*Virtue2.* Individuals are under a duty of virtue to respect the legal rights by which the state determines and protects their external freedom.

I should mention another approach to the Right/Virtue relations. For *strict* separationists such as Thomas Pogge's Kant,[14] the violation of external freedom does not involve any distinct moral wrong. Strict separationism denies Virtue2: one's self-mastery has no normative standing at all, and individuals are under no duty of virtue to respect it as such. Indeed, according to Pogge's Kant, the *law* should determine and protect bodily and property rights, but in pursuing obligatory ends, individuals have no *moral* reason to respect these rights. In cases where morality and the optimal law conflict, there is no reason to respect legal rights. Thus, while all separationists find the attempt to derive Right from Virtue misguided, some of them, viz., strict separationists, argue that that external freedom is morally worthless.[15] I will not discuss this view.

## 4. A Test-Case: Preventing an Impermissible Suicide

I am in a position to illustrate the separationist irreducibility of Virtue and Right. Consider the approach of standard (and strict) separationism to the duty to prevent impermissible suicide. As a preliminary, here is Kant on suicide:

Man cannot renounce his personality as long as he is a subject of duty, and hence so long as he lives. It is a contradiction that he should have the moral title to withdraw from all obligation, that is freely to act as if he needed no moral title for this action. To destroy the subject of morality in one's own person is to root out the existence of morality itself from the world, so far as this is in one's power; and yet morality is an end-in-itself. Thus, to dispose of oneself as a mere means

---

12    I rely on the introductory remarks offered in (Rawls 2000, at p. 177 and especially, at pp. 187 ff.).

13    Robert Pippin coins the term, "separationists" in his "Mine and Thine?" at p. 423. He distinguishes between the separationist readings of Kant's political philosophy and the reading that "derivationists" advance. See references in Pippin, ibid., endnotes 14–16.

14    Elaborated in (Pogge 2002).

15    For a somewhat different version of strict separationism, see (Willaschek 2009, 2002).

to an arbitrary end [an end of natural inclination] is to abase humanity in one's own person (homo noumenon), which was yet entrusted to an as being in the world of nature (homo phenomenon) for its preservation. (Quoted in Rawls 2000, pp. 192–93)

I will use David Velleman's reconstruction of this argument. Velleman insists that a person's value as a person "is not just his affair. . . . his value as a person inheres in him among other persons. It's a value that he possesses by virtue of being one of us, and the value of being one of us is not his alone to assess or defend. The value of being a person is therefore something larger than any particular person who embodies it." (Velleman 1999, p. 618). Imagine a person who smokes, knowing for sure that this practice will shorten his life. He wholeheartedly prefers enjoyment to longevity. Velleman argues that the smoker treats his personhood as an asset, and thus fails to fulfil the duty to respect the value of personhood.

Suicide is permissible in some cases, though. "While Kant's doctrine excludes suicide for reasons based solely on our natural inclinations, it is not always forbidden whatever the reasons. What is required are very strong reasons based on obligatory ends, which may conflict in particular circumstances." (Rawls 2000, p. 193). Consider a Francis Kamm example, in which "life involves such unbearable pain that one's whole life is focused on that pain." (Quoted in Velleman 1999, p. 618). In such a case, Velleman argues, "the pain is more than painful, since it not only hurts the patient but also becomes the sole focus of his life. Pain that tyrannizes the patient in this fashion undermines his rational agency, by preventing him from choosing any ends for himself other than relief. It reduces the patient to the psychological hedonist's image of a person—a pleasure-seeking, pain-fleeing animal—which is undignified indeed. . . . [T]his severely reduced condition of the patient can be ended only by his death." (Velleman, ibid.). I will call this the "Kamm/Velleman case".

Is there a duty of virtue to prevent immoral suicide? Does the duty to respect the value of personhood entail a duty to prevent such suicide? Is it an obligatory end? Separationism offers a suggestive answer. Virtue1 suggests that you are allowed to challenge one's decision to commit an impermissible suicide. Moreover, the fact that our personhood is shared, and that personhood should not be treated merely as means to a further end, implies that the attempt to convince the smoker to stop smoking involves no paternalism, since the preventer aims to protect the humanity that she shares with the smoker.

Is it morally permissible/obligatory to prevent an impermissible self-destructing behavior "by force"—that is, by hindering the freedom of the smoker and subjecting her to your choice? Velleman's response is complicated: "I don't go around snatching cigarettes out of people's mouths. And I'm not sure that I would forcibly try to stop someone from committing suicide solely because it would be immorally self-destructive. The impermissibility of someone else's conduct doesn't necessarily give me permission to interfere with it." (Velleman 1999, p. 614). For Velleman, there is a weighty pro tanto reason not to interfere with one's body and property in preventing one's self-destructing behavior. Use of force is presumptively wrong even in such circumstances.[16]

Separationists would analyze Velleman's intuitions by appealing to the value of external freedom, as it is embodied in Virtue2. True, preventing an impermissible suicide is an obligatory end. Yet, at least pro tanto, this end should not be promoted by interfering with the body of others. That is, you have a weighty pro tanto reason against preventing an impermissible suicide by force, even if force is necessary. The maxim that commands respect of property and bodily rights, justified by the intrinsic irreducible value of external freedom, speaks against hindering the freedom of a person in promoting an obligatory end.

Notably, in rejecting Virtue2, the strict separationism Pogge attributes to Kant rejects the intuition that hindering one's external freedom is intrinsically wrong and that therefore,

---

16    Ripstein's Kant explains the smoker's right against your intervention as follows: "the ethical argument against suicide has no bearing on rights, since it concerns only the relation between the end to be pursued and the means being used in pursuit of it. . . . the relation between an agent's ends and the means he or she uses doesn't matter for right; only the form of interaction with others does." (Ripstein 2009, p. 143).

the fact that one can prevent an impermissible suicide only by force does not entail that the prevention is pro tanto impermissible. This, however, is not the end of the story that strict separationism would tell. Non-consequentialist morality provides a different pro tanto reason against forcing people to fulfil their moral duties. All coercers can do is cause the coerced agent to act *in accordance with* the duty in question. If coercers are successful, the coerced agent did the right thing for the wrong reason; he failed to act *out of duty*.[17] I will describe such a coercion as an instance of (pro tanto wrongful) paternalism. Thus, for standard separationists, in the smoker case that concerns Velleman, use of force has two different wrong-making features, viz., it violates the smoker's external freedom, and it is paternalistic.

Consider another case. Suppose John is determined to end his life since the pains from which he suffers are unbearable. He knows that these pains are temporary, but this piece of knowledge does not change his mind. He wants to rid himself of the pains now, and self-termination is necessary for achieving this end. You can prevent John's suicide, either by temporarily locking him in a cell, or by forcibly injecting him with psychiatric medications that would "cool him down". The Virtue propositions entail (first) that preventing his suicide should be your goal, and that the necessary means for the achievement of this end, viz., hindering John's freedom, is morally objectionable. By locking John in a cell, you violate his self-mastery. As I have just noted, a different pro tanto reason against your (and the state's) interference with John's body suggests itself. By causing John to act in accordance with the moral obligation to which he is subject, the coercer does not cause him to act out of respect for morality. John is forced to *conform* with the moral reasons that apply to him, rather than to *comply* with them.

In what follows I will be assuming, like standard separationism and *pace* strict separationism, that interfering with one's body or property is morally objectionable, even if it involves no paternalism.[18]

## 5. An Objection to Separationism

This section exposes a problem in standard separationism, which the view I call "reduced separationism" (developed in the next section) aims to resolve. Consider an instance of the Kamm/Velleman case, where, arguably, although your intervention involves force and coercion it involves no paternalism. Bob intends to commit suicide because he suffers continuous unbearable pains and thinks that he is about to die very soon anyway. Bob's suicide is *subjectively* justified; understatedly, he takes it that he is about to irreversibly lose his rational nature. But Bob's factual beliefs are mistaken. He can afford an effective painkiller and a medication that would considerably improve his condition.

To convince Bob that his suicide is unjustified by his own lights, it is necessary to lock him in a cell for a while, to use his money to buy the painkiller, and to inject him with it. That is, demonstrating to him that the moral principles by which he lives speak against the suicide he plans involves hindering his external freedom. You do so, believing that you are fulfilling your duty to rescue Bob. Thanks to your intervention, Bob's pains are much less severe; consequently, he realizes that his own moral commitments rule out suicide. Contrary to the John case, in which the coercer forced John to conform with the reasons that apply to him, in the Bob version, you assist him to comply with these reasons.[19] In

---

[17] Consider Kant's famous hypothetical concerning the shopkeeper who passes up the chance to shortchange a customer only because his business might suffer if other customers found out (Kant 1964, pp. 68–69). A widely accepted conviction is that shopkeeper's action has no moral worth, because he did the right thing for the wrong reason; he acted according to a duty, not out of duty.

[18] On paternalism according to Kant, see (Wood 2002).

[19] It does not follow from my interpretation of Kantian morality that Bob's suicide is morally impermissible. An action is either moral or immoral based on the reasons for which the actor acted. And, given his factual beliefs, Bob's reasons for committing suicide are morally cogent. Instead, the reading I adopt implies that, as part of your duty to aid others, you ought to aid him to see things as they really are, even if his mistaken beliefs justify his suicide.

locking him in a cell, you merely remove the obstacle that prevented him from seeing what the maxim he holds really commands.

Some readers might suspect that this analysis, which sees no paternalism in forcing Bob to realize what he wants, illegitimately attributes to Bob false consciousness. At the time of your interference with Bob's body and property, you have no direct access to Bob's mind, so you act based on a conjecture as to what he "really" wants. The objector maintains that such an interference is paternalistic. In response, I admit that you might be wrong as to the moral principles to which Bob is committed. And, if you are wrong, your interference is paternalistic. Nevertheless, contrary to the usual false consciousness (illiberal) arguments, the analysis I offered above does not support locking Bob "up until he realizes what his true desires are". Rather, based on your acquaintance with Bob and the preferences he already expressed, you are certain that locking Bob up is a means by which you remove an obstacle that prevents him from seeing what the principles to which he is already committed command in the circumstances that he encounters. Crucially, the only grounds for your certainty are Bob's expressed, informed, and freely adopted preferences to which you were exposed just before his pains became unbearable.

How would separationism treat the Bob case? Consider Virtue1 and Virtue2, first. Together, they imply that the Bob version involves a typical conflict between two duties, the pro tanto duty to rescue Bob on the one hand, and the pro tanto duty not to hinder Bob's external freedom, on the other. Pro tanto, you ought to lock him down despite his protests because it is a necessary means for rescuing him. The prevention of his suicide is an obligatory end you ought to pursue. Pro tanto, you ought to respect his independence from you since Bob's self-mastery is intrinsically valuable. Supposedly, for separationists, the former duty, the duty to rescue Bob, outweighs the latter duty, the duty to respect Bob's private sovereignty.

So far so good: the Virtue propositions that compose separationism seem to yield a plausible result. The weakness that threatens separationism relates to the Right proposition. In adopting Right, standard separationism says that a state should protect citizens' right not to pursue obligatory ends. It follows that since locking Bob up and injecting a painkiller in him is a violation of his self-mastery, the law ought to forbid it. The fact that you are under a moral duty to assist Bob does not exempt the law from subjecting you to a legal duty to respect his property and bodily rights. This result seems counterintuitive. Moreover, it seems inconsistent with the way the law protects self-mastery in other contexts.

To see the inconsistency between the actual law and separationist's analysis of the optimal law (as I explained above using the Bob case), compare the way the common law treats the duty to rescue to the way it handles altruistic interference with one's body and property. Famously, the common law does not sanction failure to rescue an innocent person. The common law respects the external freedom of the potential rescuer, allowing her to conduct her life as she sees fit. Surprisingly, however, the common law allows an agent (the "gestor") to act on behalf and for the benefit of a principal (the "negotii"), without the latter's consent or authorization. So, suppose Bob's pains are so severe, that although he resents your interference with his body and property, he could not express his resentment. Then, according to the accepted doctrine, you could permissibly rescue Bob by locking him up *and* break his arm in doing so. If you take it upon yourself to save someone, you can act on his behalf, in light of your view about what is best for him. Moreover, Bob ought to compensate you for acting on his behalf. If you purchased the painkiller with your own money, then Bob is bound to indemnify you for your expenses. If he refuses, there is unjust enrichment, and you possess a claim to bring an action for restitution.[20]

According to another famous legal doctrine, the law allows you to use and to damage another person's property in cases of necessity and duress. You can break into Bob's cabin

---

[20] As the law is standardly understood, it does not allow intervention in a case where Bob explicitly opposes your intervention. But I am not sure this is consistent with the permission to act on Bob's behalf. After all, his protests are based on factually mistaken beliefs. For a detailed discussion see (Kortmann 2005, especially, chps. 8 and 9).

if a bear would otherwise kill you. Now, Ripstein argues that this doctrine is consistent with Right since you should compensate the victim for this violation: "you are liable in tort for the damage you cause, because it is still something you did to someone else's property." (Ripstein 2009, p. 131). The right of a person in peril "to freely use the property of another in order to preserve his own life or the lives of others," ... "always include[s] a duty on the part of the person using or destroying the property of another to compensate the owner." Indeed, "when the emergency is over, the person must pay for the property used or destroyed." (Ibid., at p. 275). However, despite Ripstein's argument to the contrary, the doctrine of necessity still seems inconsistent with Right. After all, you are liable for any damage that your negligent behavior caused. It does not follow that your negligence was legal.

In summary, two questions are left open. First, would Right render rescuing Bob in the Bob case illegal? The answer seems positive, but a positive answer seems implausible. Second, why does actual law not prohibit using things that belong to another person for a purpose that she did not set herself, such as rescuing her and rescuing oneself?

## 6. Reduced Separationism

While other answers to these questions have been offered in the literature, it seems that the most attractive one—the one offered by reduced separationism—has been under-explored. Reduced separationism advances a modification of Right:

> *Right\**. The state ought to determine the scope of the bodily and property rights of its citizens and to protect them.

> The state is entitled to hinder private freedom only for advancing its obligatory ends as a public entity, viz., securing freedom and just distribution of the means to realize it. It must be neutral with respect to the ends that its citizens pursue.

> States and individuals are entitled to interfere with citizens' body and property, *if presumably, citizens voluntarily accept a mutually beneficial and fair social rule* that allows the interference in question.[21]

I suggest that the power to redraw our personal domain by voluntary accepting social rules and the power to redraw our personal domain by consent and contracts are one and the same legal power. Evidently, I can confer on you a right to enter my house by voluntarily inviting you in. I suggest that, likewise, Bob might confer on you a right to enter his house by voluntarily accepting a social rule that allows rescuers to use the victim's property on the victim's behalf. The legal power to invite you in my house is an aspect of the nature of the rights that define my liberty in all Kantian political philosophies mentioned above. My (probably controversial) suggestion is that, in some circumstances, Kantian political philosophy should understand the voluntary acceptance of social rules as an instantiation of this very legal power.

To see the advantages of Right* consider another version of the Kamm/Velleman case. Alice suffers awful pains. She wants to stop them by committing suicide. Yet, unlike Bob, Alice does not *really* want to die. That is, she does not want to be moved by her first-order self-destructive desire. She forms a (conflicting) desire to stay alive, and a second order desire to be moved by this latter desire. Alas, the pains are awful, and the desire to stop them by suicide is compulsive. While she does not want to be moved by the self-destructive desire, this desire is too strong, and she cannot overcome its motivating force by herself.[22] To overcome the compulsive desire, Alice needs an external help, which you can provide by

---

[21] The rules should be mutually beneficial and fair, as otherwise, we should not presume that citizens *voluntarily* accept the interferences that they allow. In other words, the presumption that the acceptance of legal rule is free is justified only if the rule in question is mutually beneficial and fair. For a similar move see (Benbaji and Statman 2019, chp. 2), where we argue that if the rules of war are mutually beneficial and fair, it is presumably true that soldiers freely accept them.

[22] I employ the conceptual framework that Harry Frankfurt elaborates in his "Freedom of the Will and the Concept of Person", (Frankfurt 1971).

giving her a rare painkiller. To hand the painkiller to her, you must walk through her land. Due to her condition, Alice cannot authorize you to do so. In aiding her to overcome the self-destructive desire that tyrannizes her mental life, you must interfere with her property.

If the following factual conditions are met, Right* entails that your use of her land is not a trespass. Suppose that the accepted social norm says that, if necessary, third parties are allowed to use one's property in meeting one's basic needs. Suppose that presumably, Alice *voluntarily* accepts this rule, and that the rule is in fact, mutually beneficial and fair. If so, presumably, she authorizes you to use her land to reach her. Similarly, suppose that the social convention in our society is that friends and acquaintances kiss each other when they meet. Intuitively, if we are friends, you are entitled to presume that I allow you to kiss me in such circumstances, and that this practice involves no injustice or exploitation. Right* suggests, then, the notion of consent and authorization that figures in the legal conceptions of external freedom is sensitive to the expectations that voluntary social rules generate.

Reduced separationism replaces Right with Right*. To see the fundamental difference between standard (Right-based) and reduced (Right*-based) separationism, compare two societies, S1 and S2. In S1, the social rules that govern the various interactions among private persons are *all* Right based. Hence, the S1 rules that private persons accept and expect others to respect are neutral with respect to the ends and the life plans of citizens. The S1 legal system determines the scope of citizens' liberty, rules for correcting the violations of their external freedom, and rules of distribution of the means by which external freedom is realized. There is nothing else.

The S2 legal system and the S1 legal system are almost identical. The S2 citizens generally respect the external freedom of each other, as it is defined by the S1 legal system. Yet, their interactions are also governed by legal rules that obligate them to pursue some obligatory ends, and legal rules that allow them to do so, by interfering with others' bodies and properties in exceptional circumstances. In particular, the maxim from which you acted in helping Bob and Alice follows from a customary legal rule, which allows interfering with another person's body and property in cases of necessity and duress. Crucially, these social rules gained legal standing in S2 because people used to follow them. The rule you followed in the Alice and Bob cases became a *customary law*, viz., written, an enforceable legal rule whose causal source is informal, non-legal social rules that emerged from the bottom up, out of citizens' free and informed consent and mutual expectations.

Counterintuitively, standard separationists render the S1 regime superior to the one by which S2 is governed. This is because, S2's legal system entitles citizens to hinder others' freedom, based on their view as to the (actually) shared ends that they all ought to pursue. Standard separationists stress that a political society ought to be neutral, viz., it ought not to shape its legal system based on moral ("metaphysical") truths about which ends we should set for ourselves. States ought to concern themselves only with securing the means that will enable citizens to realize their comprehensive doctrines and with a just distribution of these means.

Much more plausibly, "reduced separationism" renders S2 superior to S1. Like separationism, reduced separationism argues for a moral duty to establish a state that determines and protects citizens' basic liberties and secures their fair share of the resources that enable them to lead a meaningful life. Also like separationism, reduced separationism asserts that the duty to establish a political society is not grounded in any comprehensive doctrine about the moral or the metaphysical truth.

Nevertheless, reduced separationism maintains that the S2 legal system is superior to S1's because of the voluntary legal rules that allow interference with Bob's and Alice's bodies and properties, despite the fact that these rules compromise the state's neutrality. Reduced separationism further maintains that Right* follows from Right. That is, the voluntariness of rules that allow interfering with one's body and property renders the S2 legal system consistent with Right. By voluntarily accepting the rules that impose obligatory ends, citizens waive some of the rights that compose their independence from each other, and thus redraw the scope of their self-mastery. Under this interpretation of

the liberties that a neutral state secures, the voluntary acceptance of legal rules is morally effective, just like a contract that changes the set of property rights of the parties. By accepting these rules, citizens allow others to interfere with their body and property in circumstances in which the legal rules allow it. For reduced separationism, citizens have a moral power to modify the scope of the rights that compose their self-mastery, by freely accepting rules that were integrated into the legal system (to which they are subject).

Thus, it would be wrong to force individuals who belong to S1 to accept the S2 legal system. Use of force involves a violation of their right to adopt a set of ends, or a life plan, independently of others' comprehensive doctrines. If individuals who belong to S1 are religious deniers, who see no value in rationality and autonomy, it would be wrong to force them to accept the S2 regime. Similarly, if S1 is a community of libertarians, who deny that they ought to rescue strangers even if the rescue is costless, or a community of Hobbesians who find altruism stupid, it would be wrong to legislate a rule that sanctions their egoism.

To see this point more clearly, compare S2 to S3. Suppose that both societies are governed by identical legal systems. They both enforce a set of legal rules that protect external freedom and promote distributive justice on the one hand, but allow interfering with one's property and body in exceptional circumstances (like the Bob case), on the other. There is a crucial difference between S2 and S3, though. In S3, rather than being customary laws, the legal rules that allow interfering with one's body and property did not emerge from the bottom up. Rather, a minority of Kantians forced a majority of religious deniers to accept them, from the top down. Members of S3 reluctantly accept rules that command pursuing obligatory ends; they do so because they fear the sanctions that back these legal rules.

Reduced separationism insists that this difference between S2 and S3 makes a fundamental moral difference. The legal system that governs S2 is consistent with Right, while the one that governs S3 is not. The S2 legal system allows interfering with citizens' bodies and properties in pursuing obligatory ends, precisely because members of S2 wholeheartedly support the rules that allow it in exceptional circumstances. In contrast, S3 forces citizens to allow the violation of their external freedom. Thus, if Bob or Alice belong to S2, then, presumably, they willingly and freely accept the customary law that allows you to use their property on their behalf; if they belong to S3, they probably accept such a rule out of fear.

Reduced separationism follows standard separationists such as Ripstein's Kant, Rawls, and Nozick by allowing coercion in establishing and maintaining the civic condition. It allows the public to use force in protecting and enhancing private freedom and distributive justice. It asserts that the public is *not* entitled to force individuals to allow the pursuit of obligatory ends if it involves interfering with the body and property of citizens. This is because the state has no normative power to allow freedom-unrelated violations of external freedom, without the actual consent of its constituents to social rules that allow them. However, unlike standard separationism, reduced separationism renders citizens' actual acceptance of mutually beneficial and fair legal rules morally effective; further, it takes actual free consent to be a valid option that citizens can realize by voluntarily following social rules.[23]

Reduced separationism is superior to standard separationism for another distinct reason. It relaxes the tension between the values which, according to the Virtue propositions, individuals ought to respect. Personhood, as well as external freedom, are intrinsically

---

23  These thinkers seem to be committed to a kind of conceptual truth: states' actions involve force and coercion. As John Simmons observes, Rawls assumes that essentially coercive political societies are a given: "we come to be within [a political society] and we do not, and indeed, cannot, enter or leave it voluntarily" (Rawls [1993] 2005, p. 136). The aim of a theory of justice according to Rawls is to describe an ideal society, viz., a society that would come "as close as a society can to being a voluntary scheme" (Rawls 1971, p. 13; quoted in Simmons 2016, p. 28). Nozick distinguishes between the minimal state on the one hand, and the dominant protective association and the ultraminimal state in the same way: the minimal state forces citizens to pay for its services and offers its services to those who cannot pay for them (Nozick 1974, pp. 22–28). Ripstein's Kant makes the same point: "the point of the contract argument is not to represent the state as the product of voluntary agreement between private wills, but to show the normative structure through which the exercise of public power is consistent with individual freedom." (Ripstein 2009, p. 199).

valuable. As the Alice and Bob cases show, the value of personhood and the value of political independence might compete. You might encounter circumstances in which you have to compromise one value in promoting the other. In S2, the Bob case involves no dilemma since, presumably, individuals willingly accept a rule that allows hindering their external freedom in the circumstances you encounter.

Thus, according to the separationist line offered here, just as individuals in the state of nature ought to exit it and to establish an S1-like neutral society, individuals in S1 ought to reform its legal regime from the bottom up. Kantian philosophy argues that the state of nature is defective because there is no regime that determines individuals' rights there, and no juridical protection of their holdings. In the state of nature, individuals would fail to handle the distinctive incompatibility relations between rational beings that occupy space. I suggest that similarly, the S1 legal regime is defective because, in some circumstances, morality offers a pro tanto reason to hinder freedom, even though external freedom is intrinsically valuable. Since individuals are subject to a duty to respect others' independence, as well as to the wide duty to respect personhood, they should try to develop an S2-like legal regime out of the S1 legal regime.

Of course, reduced separationism draws a crucial distinction between the duty to exit the state of nature and to maintain an S1-like civic condition, and the duty to reform the S1 regime towards the S2 regime. Stateless individuals might force a civic condition on those who prefer to stay in the state of nature. In effect, individuals might permissibly be "forced to be free". Moreover, the state has a normative power to use its force—to hinder citizens' freedom—in protecting and enhancing their freedom. That is, the rights that Right secures need no popular approval—they can be enforced from the top down. In contrast, rules that allow private people to hinder others' external freedom are perfectly legitimate only if all citizens freely accept them. And, clearly, perfect legitimacy is an unachievable ideal.

Is the S2-legal system neutral? No. The S2 regime is based on a comprehensive moral doctrine, according to which John, Bob, and Alice should not commit suicide and, furthermore, preventing their suicide is an obligatory end. Being consistent with Right, in S2, lack of neutrality is justified. Contrary to the pluralistic society that Rawls imagines (S1), individuals living in S2 are not religious deniers. They are Kantians who share the ends that, according to non-consequentialist morality, they ought to pursue. The S2 legal system is legitimate only because S2 happens to be more homogenous than the society for which Rawls designed his theory of justice. Thus, reduced separationism agrees with Rawls' political liberalism that, in a more pluralistic society like S1, enforcing S2's legal system is wrongful because it interferes with the body and property of individuals, without their consent. (It opposes coercing Kantian morality for another reason: forcing the S2 legal system from the top down is paternalistic.)

Two objections merit attention. It might be asked, why would S2 need a Right-based legal sub-system at all? Why do members of such a society not agree on a legal system that protects humanity rather than external freedom? Are Right-based rules not superfluous in a homogenous community of Kantians who share a comprehensive doctrine that centralizes the idea of humanity? No. Like standard separationism, reduced separationism insists that external freedom is intrinsically valuable, and as such, should be treated as an end-in-itself. A failure to legislate a Right-based legal system is a moral failure.

Secondly, it might be objected that in a society of Kantians such as S2, the S1 legal regime would have the same impact as the S2 legal regime. In such a homogenous society, there would be no difference between a legal regime composed of rules that protect only external freedom and a legal regime that introduces laws that entitles you to aid Bob by infringing with his bodily and property rights. To see why, imagine a hungry child who steals your apple. As a Kantian you do not to take legal action against her since, as a Kantian, you believe that you ought to assist her rather than sue her. Similarly, if you and Bob live in a society of Kantians, Bob would not take legal action against you, whether this society is governed by an S1 legal regime, or by an S2 legal regime. He would realize that locking him up was morally permissible as a means for achieving an obligatory end (viz.,

saving him from death). But this objection ignores the aim of the S2 regime: a *legal* right to interfere with Bob's body and property. S2 denies Bob normative power to take legal action against you, and denies the state a right to sanction your hindrance of Bob's freedom. More importantly, thanks to the legal regime that governs S2, you are under no duty of virtue not to interfere with Bob's body and property. Indeed, even if S1 is a society of Kantians, and no legal action would be taken against you in the Bob case, S1 differs from S2 in a morally important way: only in S2 did you not compromise the value of Bob's external freedom.

## 7. Conclusions

The separationist interpretation of the Virtue/Right distinction advanced here has three components. First, non-consequentialist morality suggests that there might be cases where, pro tanto, one ought to promote obligatory ends by hindering another person's freedom. Second, in those circumstances as well, there might be a weighty moral reason against hindering the self-mastery of this person. Third, in a perfectly just society, the legal rules that individuals willingly accept would allow interference with another person's body and property in such cases. In a perfectly just society, the interference would involve no violation of the rights that constitute self-mastery.

According to the Kantian approach that reduced separationism offers, external freedom and humanity are both intrinsically valuable. Expectedly, in certain cases, these values compete; in promoting obligatory ends dictated by the intrinsic value of personhood, hindering external freedom might be necessary. A legal system that fulfils its duty to determine, protect, and enhance external freedom would prohibit such violations, and, as such, would neglect the intrinsic value of humanity. Yet, in a pluralistic society where, say, secular humanists and religious deniers live together, this neglect is justified. A legal system has no moral power to hinder the freedom of its subjects (who believe humanity to be worthless) in order to make them respect personhood.

Thus, in cases in which pursuing obligatory ends requires hindering the freedom of others, virtuous private people face a dilemma. Reduced separationism renders the very possibility of such a dilemma a defect. Accordingly, it suggests that the dilemma can be resolved only if (by exercising *no* force or paternalism) Kantian humanists convince religious deniers to willingly convert to Kantianism. Reduced separationism further suggests that in a society of Kantians, where the rules that allow violating external freedom in special circumstances are voluntarily accepted, citizens redraw the scope of their external freedom. Interfering with one's body and property in pursuing obligatory ends might involve no violation of external freedom.

**Funding:** This research was funded by [The Israeli Research Foundation] grant number [396/18].

**Conflicts of Interest:** The author declares no conflict of interest.

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
