# Peer review of "Morality, Voluntary Laws, and State Neutrality"

_laws, 2022_

Round 1

Reviewer 1 Report

I enjoyed reading a novel deeper take on a familiar topic. I must commend the author for selecting the theme and the presented arguments, which are sound, coherent and clear to the reader.

Nevertheless, I have specific proposals for improvement that I believe should be addressed, before the article is published.

1. I am unsure if there is a problem in the version of the manuscript that I am reading or some technical issue, but the list of literature/sources is lacking. While looking at the footnotes, everything seems to be cited appropriately; it is necessary to see the list of references and their congruence with the footnotes to ensure that the referencing is in order.

2. The methodology used by the author seems appropriate and customary for the field. Nevertheless, it would benefit the reader if, in the introduction, the methods used were described in a paragraph or so.

3. There should be a more systematic engagement with existing literature, which could be a separate subheading. Still, it would entirely suffice if the author dedicated a paragraph or two in the introduction to even more clearly describe how their research fits into the state-of-the-art and how it goes beyond it.

Taking into account these considerations, the article is interesting and warrants publication.

Author Response

I thank the reviewer for their wonderful comments. Here are my responses. 

  1. I did not include  list of literature/sources , I will do so, once the paper is finally accepted. 
  2. Analytical philosophy has no "method", I think.
  3. I omitted the material on the justice/truth distinction.  I think that the existing literature on the other themes discussed in the paper is properly presented.  

Author Response

I thank the reviewer for great comments. Here are some responses:

 1a. The reviewer offers a nice summary of the essay; this somehow undermines their claim that the essay is hard to follow. 

1b. I use the distinction between duties of right and duties of virtue, employed by Ripstein's Kant.

1c. Thanks!! I agree, and omitted the discussion on truth and justice. 

2. I am sorry, but I don't understand this remark. 

3. Thanks for this comment. I address it in fn. 54.

4. I fail to understand this comment as well. 

5. As I emphasize in the text, Ripstein, Pogge and Pippin rejects the idea that political philosophy is an implication of moral philosophy. True, the early Rawls anchored the first principle of justice and its priority in Kantian morality. In political liberalism he abandons this idea. 

5a. I agree that there are many differences between Ripstein Rawls and Nozick, and re-emphasize this in the modified version. Thanks!

Round 2

Reviewer 2 Report

Please see attached file. Just in case, I include comments below.

The author made a good faith effort to address my concerns. As a result, I recommend that the article be published. At this point, I would not require that the author address the clarified concerns regarding points (2) and (4).

Here are the clarified concerns.

(2)   Hohfeldian Powers. It is unclear whether Reduced Separationism is Separationism plus a theory of moral Hohfeldian powers or something more fundamental.

Reduced Separationism says the following.

·       Right*. The state ought to determine the scope of the bodily and property rights of its citizens and to protect them. The state is entitled to hinder private freedom only for advancing its obligatory ends as a public entity, viz., securing freedom and just distribution of the means to realize it. It must be neutral with respect to the ends that its citizens pursue. States and individuals are entitled to interfere with citizens' body and property, if presumably, citizens voluntarily accept a mutually beneficial and fair social rule that allows the interference in question.

The concern is the role that mutually beneficial and fair social rules play. Is the issue that whatever the citizens voluntarily accept allows the state to interfere with citizens’ body and property, regardless of whether the interference is beneficial and fair? Alternatively, is the issue of what interferences with citizens’ body and property rights – understood as a necessary condition – along with whether the citizens voluntarily accept the interference?

(4)   Rights. It is unclear if what allows citizens to accept mutually beneficial or fair legal rules and change their moral boundaries is the nature of rights, self-ownership, or a Kantian imperative. If the nature of rights, self-ownership, or a Kantian imperative is what allows for it, then it is unclear why this is just an application of Kantian moral philosophy. On such an account, issues of Kantian political philosophy do not arise as a distinct interpretation of Kantianism.

On different accounts of the Kantian morality – including the author’s account – citizens can accept legal rules because they (a) have moral rights, (b) own themselves, or (c) the Categorical Imperative allows for it. Which justifies citizens’ power to accept legal rules? How is the answer to this previous question a distinction interpretation of Kantian morality?  

Author Response

Thanks for the clarifications. Here is what I say in essay in response to 2.

"I suggest that the power to redraw our personal domain by voluntary accepting social rules and the power to redraw our personal domain by consent and contracts are one and the same legal power. Evidently, I can confer on you a right to enter my house by voluntarily inviting you in. I suggest that, likewise, Bob might confer on you a right to enter his house by voluntarily accepting a social rule that allows rescuers to use the victim's property on the victim's behalf. The legal power to invite you in my house is an aspect of the nature of the rights that define my liberty in all Kantian political philosophies mentioned above. My (probably controversial) suggestion is that, in some circumstances, Kantian political philosophy should understand the voluntary acceptance of social rules as an instantiation of this very legal power."

In response to 4, I added fn. 45.